# BnGF14-2c Positively Regulates Flowering via the Vernalization Pathway in Semi-Winter Rapeseed

**DOI:** 10.3390/plants11172312

**Published:** 2022-09-03

**Authors:** Shihang Fan, Hongfang Liu, Jing Liu, Wei Hua, Jun Li

**Affiliations:** 1Oil Crops Research Institute of the Chinese Academy of Agricultural Sciences, Key Laboratory of Biology and Genetic Improvement of Oil Crops, Ministry of Agriculture and Rural Affairs, Wuhan 430062, China; 2Hubei Hongshan Laboratory, Wuhan 430070, China

**Keywords:** *BnGF14-2c*, overexpression, early flowering, vernalization, *Brassica napus* L.

## Abstract

14-3-3s are general regulatory factors (GF14s or GRFs) involved in a variety of physiological regulations in plants, including the control of flowering time. However, there are poorly relevant reports in rapeseed so far. In this study, we identified a homologous *14-3-3* gene *BnGF14-2c* (*AtGRF2*_Like in *Brassica napus*) in rapeseed based on bioinformatic analysis by using the sequences of the flowering-related *14-3-3s* in other plant species. Then, we found that overexpression of *BnGF14-2c* in the semi-winter rapeseed “93275” promoted flowering without vernalization. Moreover, both yeast two-hybrid and bimolecular fluorescence complementation analysis indicated that BnGF14-2c may interact with two vernalization-related flowering regulators BnFT.A02 and BnFLC.A10., respectively. qPCR analysis showed that the expression of *BnFT* (*AtFT*_Like) was increased and the expression of two selected vernalization-related genes were reduced in the overexpression transgenic plants. Further investigation on subcellular localization demonstrated that BnGF14-2c localized in the nucleus and cytoplasm. The results of RNA-seq analysis and GUS staining indicated that *BnGF14-2c* is ubiquitously expressed except for mature seed coat. In general, the interaction of 14-3-3 and FLC was firstly documented in this study, indicating BnGF14-2c may act as a positive regulator of flowering in rapeseed, which is worthy for more in-depth exploration.

## 1. Introduction

Flowering, the transition from vegetative to reproductive, as a crucial decision feature in the life cycle of higher plants, is switched by both external and internal signals [1,2,3]. To date, the underlying mechanisms of floral transition have been well-characterized in the model plant *Arabidopsis*, including photoperiod pathway, vernalization pathway, autonomous pathway, gibberellins pathway and age pathway [4,5]. These pathways act independently and involve extensive crosstalks and feedback loops with each other [6,7]. A process of low temperature induction during the vegetative growth period known as vernalization is required for flowering in many plants [8,9,10]. The vernalization-requiring monocarpic plants are called biennial or winter annual [11,12], and the rapid flowering types that can complete the life cycle in one season are named summer or spring annual [13].

In Arabidopsis, a series of genes have been illuminated to be related to the vernalization pathway, such as FLC (FLOWERING LOCUS C), FRI (FRIGIDA), VRN1 (REDUCED VERNALIZATION RESPONSE 1), VRN2 (REDUCED VERNALIZATION RESPONSE 2), VRN5 (VERNALIZATION 5), VIN3 (VERNALIZATION INSENSITIVE 3) and so on [14,15,16]. FLC and FRI are two dominant genes, which cooperate to complete the dependence of flowering on vernalization. FLC acts as a dose-dependent inhibitor of flowering, suppressing the FT (FLOWERING LOCUS T), which encodes a florigenic protein, as well as inhibiting the FD (FLOWERING LOCUS D) and SOC1 (SUPPRESSOR OF OVEREXPRESSION OF CO 1) genes in SAM (Shoot Apical Meristem) [17,18,19,20]. FRI is a positive regulator of FLC to inhibit flowering and the FRI mutation can lead to flowering prematurely in Arabidopsis [21,22,23]. VRN1, VRN2, VRN5, and VIN3 belong to the negative regulators of FLC [24,25,26,27]. Its inducibility guarantees that the expression of FLC is reset during the embryonic developmental stage of offspring to ensure the regain of vernalization [28,29,30].

The general regulatory factors 14-3-3s (named as GF14s or GRFs) [31,32,33], are highly conserved and ubiquitously expressed in eukaryotes. These proteins can form dimers and have nine antiparallel *α*-helices [33]. These structures facilitate the binding of interacting proteins, and bring two or more proteins together to form multiprotein complexes [34,35,36]. In plants, 14-3-3s regulate a diverse set of cellular and physiological processes, such as responding to biotic or abiotic stress [37,38], hormone signaling [39,40], regulating tissue development [41] and affecting the formation of some important agronomic traits [42] in various crops. Previous studies suggested that 14-3-3s are involved in the flowering regulation in several plant species. For example, overexpression of the tomato *14-3-3* gene compensates the early flowering phenotype of the *sp* mutant (*SP*, *Self*
*Pruning*) in tomato [43,44]. In bamboo, it was also found that at least three PvGF14s may serve as negative regulators of flowering by interacting with PvFT; the interactions were confirmed by Y2H (yeast two-hybrid screening) and BiFC (bimolecular fluorescence complementation) [45]. However, the *Arabidopsis* mutants of *14-3-3μ* and *14-3-3υ* flower late and exhibit long hypocotyls under red light, with little effect under blue or far-red light [46,47]. Studies in rice revealed that *RCN* (*TFL1*_Like, *TERMINAL FLOWER 1*) antagonizes *Hd3a* (*FT*_Like) in the inflorescence development by competition for complex formation with *14-3-3* and *OsFD*, which implied a molecular basis for the flowering regulation balance between FT and TFL1, and the interactions were detected by Y2H and BiFC [48,49,50,51,52,53]. In cotton, Gh14-3-3 can interact with GhFT and GhFD to generate FACs (Florigen Activation Complexes) to affect the flowering time, and the interactions were proven through Y2H, BiFC, and pull-down assay [54,55]. Overexpression of a foxtail millet *14-3-3* gene *SiGRF1* in *Arabidopsis* led to flowering earlier than that of wild type under high-salt stress [56]. Taken together, all these aforementioned reports suggested that the 14-3-3s may act as important regulators that are involved in the floral transition in different species.

Rapeseed (*Brassica napus* L.) is a major oil crop widely cultivated in several temperate regions of the world. It is adapted to different environments by modification of the flowering time and the requirements for vernalization [57]. According to the respective relationships between the flowering time and the vernalization requirements, Rapeseed cultivars can be divided into three types, winter type (only flower after exposure to an extended strong cold period) [10,58], semi-winter type (can flower with milder or no cold) [59,60,61,62] and spring type (sowing in spring or summer and harvest before the fall) [63,64,65].

Rapeseed, with three different flowering sub-patterns, has not been reported on the functions of 14-3-3 in flowering. In this study, we cloned a homologous gene of *14-3-3* (named as *BnGF14-2c*) in rapeseed. Overexpression of this gene in a semi-winter rapeseed cultivar (93275) could promote flowering without vernalization. Then, the expression of the *BnFT* and two genes (*Bn**FLC* and *Bn**FRI*) related to the vernalization pathway of flowering were checked by qPCR analysis. Moreover, the candidate interacting proteins of BnGF14-2c were detected both in vitro and in vivo. Subsequently, a series of studies including the gene expression analysis, protein subcellular localization were also conducted. In conclusion, our results provided new insights into the role of 14-3-3 in rapeseed flowering possibly via the vernalization pathway.

## 2. Results

### 2.1. Characterization and Sequence Alignment of BnGF14-2c in Rapeseed

BnGF14-2c (BnaA07g34180D, Darmor-*bzh*) is one of the 47 14-3-3 proteins in rapeseed identified by blastP and phylogenetic analysis (data were not shown). The full-length cDNA of *BnGF14-2c* is 1203 bp including a 897 bp open reading frame encoding a protein that with 299 amino acids. Multiple sequence alignment was performed with the 14-3-3 orthologs from other plants, which have been proved to be participating in the flowering regulation, and the result showed that BnGF14-2c is highly conserved with these 14-3-3s (Figure 1a). All of the sequences have the conserved core region comprising nine antiparallel *α*-helices and divergent N/C terminis. The model image of 3D matching prediction showed that BnGF14-2c could form a homologous dimer with a horseshoe shape, typically of 14-3-3 proteins (Figure 1b). Therefore, we made a bold speculation that BnGF14-2c may also function in the flowering regulation in rapeseed.

### 2.2. Overexpression of BnGF14-2c Showed an Early Flowering Phenotype without Vernalization in Semi-Winter Rapeseed

The full-length open reading frame of *BnGF14-2c* was cloned from Darmor and introduced into the semi-winter rapeseed “93275” to generate *35S::BnGF14-2c* overexpression transgenic plants. A total of 12 independent transgenic lines exhibiting herbicide resistance in the T1 generation was confirmed by PCR with the specific primers (Appendix A). Among them, four lines showed an early flowering phenotype without vernalization compared to the wild-type (WT), and two lines were selected for further analysis (OE-21^#^, OE-22^#^) (Figure 2a). Real-time quantitative PCR analysis showed that the expression levels of *BnGF14-2c* in transgenic lines were at least three-folds higher than WT (Figure 2b). Subsequent phenotypic observations found that the transgenic plants bolted and bloomed about 60 days after sowing, while wild-type plants had to be delayed for a long time (more than 170 days) to partially enter into the reproductive growth stage (some individual wild-type plants even took longer to flowering or did not bloom at all in the greenhouse) (Figure 2a,c). In addition, we found that the number of rosette leaves (when the first floral bud was visible) of wild-type was more than twice as much as the transgenic lines (Figure 2d).

### 2.3. BnGF14-2c May Interact with FT and FLC Respectively Both In Vitro and In Vivo

Previous studies have indicated that FT or FT_Like proteins can interact with 14-3-3 isoforms affecting flowering time in many plant species, and the interactions were observed both in the cytoplasm and the nuclei [48,49,50,51,52], but the FLC protein has not been reported to interact with 14-3-3s. As a heterotetraploid plant species, rapeseed has a high copy number of homologous genes, and these genes maybe subfunctionalized to different degrees. Previous reports have been identified that *BnFT.A02* as a candidate gene for flowering in rapeseed [66,67,68], *BnFLC.A02* and *BnFLC.A10* were two floral integrator genes with the vernalizing conditions in winter rapeseed [15,69]. To testify the possible role of *BnGF14-2c* in the vernalization pathway, the florigen gene (*BnFT.A02*) [68] and a key gene in the vernalization pathway (*BnFLC.A10*) were selected as candidate interacting proteins [58,70].

Y2H was performed by cloning the coding sequences of *BnGF14-2c* into the prey vector pGBKT7, while *BnFT.A02* and *BnFLC.A10* into the bait vector pGADT7 (Figure 3). The plaque of SD/-LT+X (SD/-Leu/-Trp/X-*α*-gal) were all in good condition, indicating that the co-transformation of the prey vector and the bait vector was successful. The positive control grew well on the SD/-LTHA+X (SD/-Leu/-Trp/-His/-Ade/X-*α*-gal) medium no matter with or without 3-AT, while the negative control group was inhibited seriously. The auto-activation verification group could grow on the SD/-LTHA+X without 3-AT, but the plaque turned red and the growth was inhibited. The auto-activation group did not grow after the addition of 3 mM 3-AT, indicating that BnGF14-2c had a certain degree of self-activation activity, but the activity could be inhibited under the action of 3 mM 3AT. The test group of AD:BnFT.A02 and BK:BnGF14-2c was grown well on the SD/-LTHA+X medium, even with 3 mM 3-AT, suggesting that the interaction was strong. The interaction between AD:BnFLC.A10 and BK:BnGF14-2c was weak indicated by the red plaque that grew slowly. Taken together, our results suggested that BnGF14-2c possessed the abilities of interaction with BnFT.A02 and BnFLC.A10 in vitro.

To further testify these interactions in vivo, the BiFC assays in the leaves of tobacco (*Nicotiana benthamiana*) were carried out. The fluorescence signals from NE:BnGF14-2c + CE:BnFT.A02 interaction were detected in the cytoplasm and the nucleus. The interaction of NE:BnGF14-2c + CE:BnFLC.A10 was mainly detected in the nucleus, and slightly lower in the cytoplasm, but no GFP signal was detected in the NE:BnGF14-2c + CE (Figure 4). All these results suggested that BnGF14-2c may interact with BnFT.A02 and BnFLC.A10 in plant cells in vivo.

### 2.4. BnGF14-2c Negatively Affect the Expressions of Two Vernalization-Related Genes and Positively Influence the BnFT Expression

To study the molecular mechanisms of *BnGF14-2c* in the flowering vernalization pathway, we then tested the expression levels of two vernalization-related genes (*BnFLC, BnFRI*) and a florigenic gene (*BnFT*) by qPCR. The results showed that *BnFT* expression was dramatically increased in *BnGF14-2c* overexpressing transgenic lines than that in the wild type. While, the expression levels of *BnFLC* and *BnFRI,* were decreased in those transgenic lines compared with that of wild-type plants (Figure 5). Therefore, our results implied that *BnGF14-2c* may negatively affect the expression levels of the two selected vernalization-related genes and positively influence the *BnFT* expression. These results might explain why overexpression of *BnGF14-2c* in semi-winter rapeseed promotes flowering without vernalization.

### 2.5. BnGF14-2c Is Ubiquitously Expressed

To characterize the expression patterns of *BnGF14-2c* in detail, we constructed a heat map to exhibit the expression levels of *BnGF14-2c* across tissues in two rapeseed cultivars (“ZS11” and “862”) according to the transcriptome data obtained previously in our group (Figure 6). The result indicated that *BnGF14-2c* was generally expressed at relatively high levels in most tissues and stages of these two varieties. The gene was particularly highly expressed in the stamen, stem, and flower bud.

Promoter determines the spatiotemporal pattern of gene expression and usually locates upstream of the initial codon about 1000–2000 bp in length. To check the promoter activity of *BnGF14-2c*, a 1495 bp upstream sequence of *BnGF14-2c* from the starting code was cloned as the promoter sequence and fused with the reporter gene *GUS* to construct the plant expression vector *proBnGF14-2c::GUS*, and then transformed into the wild-type *Arabidopsis* by the floral-dip method. Homozygous transgenic plants were obtained by continuous self-crossing in the T3 generation. Further GUS staining results demonstrated that GUS activity could be detected in almost all the tissues focused (seedling, leaf, root, stem, flower, and silique), except for the mature seed coat (Figure 7). All these results implied that the *BnGF14-2c* promoter may be a constituent promoter driving ubiquitously gene expression in all tissues and the *BnGF14-2c* gene may be involved in various aspects of plant growth and development.

### 2.6. Subcellular Localizations of BnGF14-2c, BnFT.A02 and BnFLC.A10

BnGF14-2c was firstly predicted to localize to the nucleus, but some studies have pointed out that the 14-3-3 may also locate in the cytoplasm. In order to verify this result, *GFP* was fused to the C-terminus of *BnGF14-2c* and the fusion construct was introduced into tobacco epidermal cells for transient expression. Fluorescence observations showed that BnGF14-2c was localized in the nucleus and cytoplasm, and the fluorescence signal was particularly strong in the nucleus (Figure 8). The subcellular localizations of BnFT.A02 and BnFLC.A10 were also detected according to the method mentioned above. Our results indicated that BnFT.A02 and BnFLC.A10 may both be localized in the nucleus and cytoplasm (Figure 4).

## 3. Discussion

Plant 14-3-3 proteins, as general regulatory factors, have been implied to be involved in plant growth/development and stress responses, especially its functions in plant flowering regulation by interacting with FT in *Arabidopsis* [46,47], with Hd3a (FT_Like) and OsFD in rice [48,49,50,51,52], with GhFT in cotton [54], with SP in tomato [43,44], and with PvFT in bamboo [45]. However, the 14-3-3 proteins’ role in rapeseed flowering has not been reported yet.

The result of multiple sequence alignment demonstrated that the isoforms of GF14s possess a conserved core region comprising nine antiparallel *α*-helices and divergent N/C termini (Figure 1a). The conserved antiparallel *α*-helices are capable of binding a separate phosphorylated target protein to participate in diverse pathways. The primary diversity occurs at the N and C termini, which are related to dimerization and target binding, respectively. Intriguingly, the N terminal sequence of BnGF14-2c is unimaginable longer than any of the other family members (Figure 1a). After careful comparison, we found that the 40th amino acid at the N-terminal may be another start codon. If removing the 39 amino acids at the N-terminal will have no effect on the conserved sequence at all, and the homodimer model was also unaffected. Based on these results, we speculated that this may be a part of UTR (Untranslated Regions) or a subfunctionalization of BnGF14-2c.

As a leading consumer of edible oil in the world, China is short of edible vegetable oil and relies heavily on imports. If the planting area of rapeseed can be expanded, this situation will be greatly alleviated. The double-cropping rice area is the area with the largest potential resources of rapeseed in China. However, after the harvest of double-cropping rice, numerous winter idle fields are abandoned due to the lack of rapeseed varieties that can be sown late and harvested early. Therefore, the selection and creation of special early-maturing rapeseed varieties with a shorter duration period can make full use of winter idle fields in double-cropping rice areas and relieve the pressure on China’s edible oil imports. Although the flowering traits are significant for rapeseed, but there is still no report about the involvement of 14-3-3 in regulating rapeseed flowering. Wild-type “93275” as a typical semi-winter rapeseed is widely adapted to the climatic conditions of the middle and lower reaches of the Yangtze River. In this study, overexpressing *BnGF14-2c* in “93,275” could promote flowering without vernalization, both in the greenhouse and field experiments (Figure 2a and Appendix A). Furthermore, the transgenic plants growing in the field showed dwarf and multi-branches with inconspicuous main stems. It is possibly due to the precocious bolting and flowering of the transgenic plants before winter comes, and then the low-temperature stress in winter prevents the growth of the main inflorescence by suppressing the apical dominance, which led to the plants to form relative more side branches in the next springtime.

Although rapeseed and *Arabidopsis* share common ancestor species, *B. napus* (*n* = 19, AA = 10, CC = 9) is an allotetraploid hybrid of *B. rapa* (AA) and *B. oleracea* (CC) under natural conditions, leading to many more gene copies within the *B. napus* by a series of whole-genome duplication. For example, the *FT* gene has six homologous in rapeseed scattering on different chromosomes, such as A2, A7, C2, C4, and C6 [68,71]. Likewise, nine copies of *BnFLCs* are located on A2, A3, A10, C2, C3, and C9 [72]. Except for the divergence of intron and promoter regions, the coding region sequences of these homologous genes are highly conserved, but the gene expression patterns show great differences [68,71,72,73]. In *Arabidopsis*, vernalization suppresses the floral repressor gene *FLC* and upregulates floral promotion genes *FT* [74]. Previous reports have identified *BnFT.A02* as a candidate gene for flowering in rapeseed [66,67,68]. The studies on vernalization requirement and response have also found that the allelic variations of *BnFLC.A02* and *BnFLC.A10* were two floral integrator genes with the vernalizing conditions in European winter rapeseed [15,69]. Here, we selected *BnFT.A02* and *BnFLC.A10* as the two candidate proteins to check their interactions with BnGF14-2c, and our results from Y2H and BiFC both verified the interactions.

The qPCR showed that the expression of *BnFT* was up-regulated and two genes (*BnFLC* and *BnFRI*) participating in the vernalization pathway were down-regulated in the rosette leaves before the bolting stage in the *BnGF14-2c* overexpressing lines (Figure 2a). Therefore, we speculated that BnGF14-2c may be involved in the fine-tuning of the rapeseed flowering via at least the vernalization pathway. Whether BnGF14-2c possibly involved in the other flowering pathways is still a question needed to be answered in the future. In conclusion, we discovered that *BnGF14-2c*, a conserved 14-3-3 gene, may act as a positive regulator of flowering in rapeseed possibly via the vernalization pathway in semi-winter rapeseed.

## 4. Materials and Methods

### 4.1. Plant Materials and Phenotypic Measurements

The semi-winter rapeseed cultivar “93275” was used as recipient material for genetic transformation. The greenhouse was under the condition (16 h, 24 °C light/8 h, 22 °C dark) with 120 μmol s^−1^ m^−2^ light intensity. The field experiments were carried out in the base (30° N, 113° E, Hanchuan, China) dedicated to the cultivation of genetically modified materials. Flowering time was measured by calculating the number of days from sowing to the first flower emergence. The number of rosette leaves was recorded on the flowering day. At least nine individual plants of each line were used for data measurement.

### 4.2. Bioinformatic Analysis

The sequences of the flowering-related *14-3-3s* in other plant species (such as *AtGF14μ/ν* in *Arabidopsis* [46,47], *LeGF14-2/5/74* in tomato [43,44], *OsGF14-c* in rice [48,49,50,51] and *Gh14-3-3* in cotton [54]) were manually extracted based on their accession numbers. Then, HMM (Hidden Markov Model) and BLASTp (E value > 10^−5^, Darmor-*bzh* V5 genome information, http://yanglab.hzau.edu.cn/BnTIR) (accessed on 5 October 2021) were applied for the identification of 14-3-3 proteins in *B. napus* according to the amino acid sequences of those flowering-related 14-3-3s. Subsequent phylogenetic analysis showed that BnGF14-2c is indeed a member of the 14-3-3 gene family in rapeseed. Then, the sequence alignments were performed by CLUSTALW bio-soft (https://npsa-prabi.ibcp.fr/cgi-bin/npsa_automat.pl?page=/NPSA/npsa_clustalw.html) (accessed on 5 October 2021). The 3D dimer model of BnGF14-2c was predicted by the online software (https://swissmodel.expasy.org/interactive) (accessed on 5 October 2021).

### 4.3. Vectors Construction and Transformation

According to the published rapeseed genomic sequence of “Dormar”, we designed the cloning primers specific for *BnGF14-2c* and obtained the sequence by PCR cloning (the primers were listed in Appendix A). The plant binary overexpression vector *35S::**BnGF14-2c* was constructed by the TOPO recombination method and pEarleyGate100 [75] as the skeleton vector. Then, the hypocotyl segments of rapeseed “93275” were impregnated by *Agrobacterium tumefaciens* to obtain transgenic plants as described previously [76].

### 4.4. RNA Extraction and Quantitative Real-Time PCR

Total RNA was extracted from the leaves at the vegetative stage of rapeseed using the Plant RNA Extraction Kit (TaKaRa, Maebashi, Japan) with PrimeScript™ RT reagent Kit (TaKaRa, Japan) that was used for cDNA synthesis. Quantitative Real-Time PCR was conducted by Hieff^®^ qPCR SYBR Green Master Mix (Low Rox Plus) according to the methods established in our previous studies [77]. Three replicates were performed for the analysis of each gene. The rapeseed *BnTMA7* gene was used as an internal control. The primers used in this study were listed in Appendix A.

### 4.5. Bimolecular Fluorescence Complementation

The sequences of *BnGF14-2c*, *BnFT,* and *BnFLC* were cloned from the leaf cDNA of Darmor. BiFC vectors (pSPYNE-35S, pSPYCE-35S) were constructed as NE-*BnGF14*, CE-*BnFT*, CE-*BnFLC* (the primers were listed in Appendix A), and these constructed vectors were transiently transformed into the tobacco leaves as described by Katia [78]. The nucleolus mCherry marker was used simultaneously and its signals were observed as mentioned above. 

### 4.6. Yeast Two-Hybrid

The yeast two-hybrid vector system pGBKT7/pGADT7 was used to test the interactions between BnGF14-2c and the other candidate proteins. The vectors were constructed as pGBKT7-*BnGF14-2c*, pGADT7-*BnFT*, and pGADT7-*BnFLC* (the primers were listed in Appendix A). The plasmids were then transformed into AH109 strain and selected on the control media (SD-Leu/-Trp/X-*α*-gal) plate by incubation for three days at 30 °C. The interactions were detected on the selective plates (SD-Leu/-Trp/-His/-Ade/X-*α*-ga). Grew on the SD-Leu/-Trp/X-*α*-gal indicative of a successful transformation with both bait and prey plasmids. The yeast plaques grow on the SD-Leu/-Trp/-His/-Ade/X-*α*-gal with the blue color indicating an interaction between both proteins. Plaque cannot grow or turn red, indicating no interaction. AD-largeT and BK-p53 were the positive control, AD-largeT and BK-laminC were the negative control, AD and BK- BnGF14-2c were the negative controls to eliminate the self-activating activity.

### 4.7. Heat Map Analysis with the Transcriptome Data of ZS11 and 862

The RNA-seq data of different tissues (ThreeLeafStage, BoltingStage, FloweringStage, and FruitingStage, details are as follows, Leaf_ThreeLeaf, Seedling, Root_ThreeLeaf, Leaf_Bolting, Root_Bolting, Pistil_Flowering, Stamen_Flowering, Petal_Flowering, FowerBuds_Flowering, Stem_Base, Stem_Top, Leaf_Flowering, AxillaryBud, CaulineLeaf_Flowering, Silique_Fruiting_3d, Silique_Fruiting_10d, Ovule_15d, SiliqueWall_15d, SiliqueWall_25d, Embryo_25d, Seed_Mature) of two rapeseed varieties (ZS11 and 862) obtained in our previous study was valuable for gene expression analysis. The transcriptome data of *BnGF14-2c* was extracted and the abundance was calculated by RPKM (Reads per Kilobase per Million Mapped Reads, value of log_2_).

### 4.8. Histochemical GUS Staining

A 1495 bp fragment upstream of the *BnGF14-2c* start codon (ATG) was cloned into binary vector pDX2181G to fuse with the *GUS* reporter gene (the primers were listed in Appendix A) and then transformed into the *Arabidopsis*. After several rounds of hygromycin selection and PCR verification, a total of 16 positive and homozygous lines were obtained. Then, the tissues (7-day-after-sowing seedling, leaf, root, stem, flower, and 15-day-after-flowering silique) were collected for GUS staining according to the method described by Jefferson [79].

### 4.9. Subcellular Localization

The plasmid pcombia3301 was used for constructing the subcellular localization vectors (*35S::BnGF14-2c-GFP*, *35S::BnFT.A02-GFP*, and *35S::BnFLC.A10-GFP*) (the primers were listed in Appendix A). FIB2-mCherry specifically expressed in the nucleolus was selected as a marker, and *35S::GFP* was used as the positive control [80]. *Agrobacterium tumefaciens* (GV3101) was used to mediate the transient transformation into epidermal cells of tobacco (*N. benthamiana*) leaves. After 3 d transformation, GFP signals were observed under a confocal laser scanning microscope (Nikon, A1).

### 4.10. Statistical Analysis

All the experiments were repeated three times. The data were analyzed with SPSS 17.0 (SPSS Inc., Chicago, IL, USA) and was presented as the mean standard error. The mRNA expression levels were analyzed with a comparative 2^−∆∆^^Ct^ method. For comparison of two groups of data, the two-sided Student’s *t*-test was used, thereinto, asterisks “*” *p* < 0.05 and “**” *p* < 0.01 was considered to be statistically significant.

## Figures and Tables

**Figure 1 plants-11-02312-f001:**
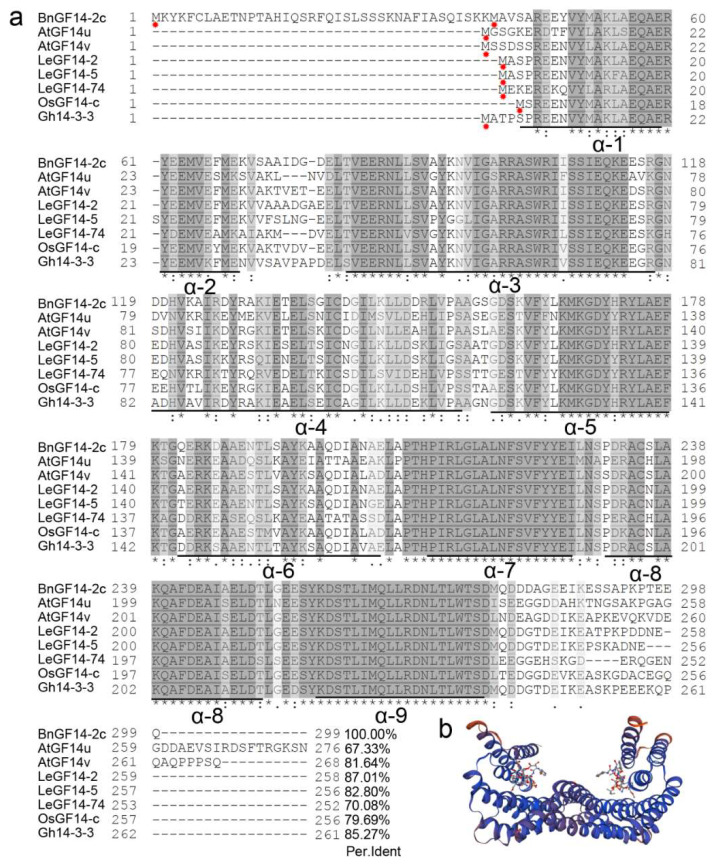
Homology alignment and homodimer 3D structure modeling of BnGF14-2c. (**a**): Homology alignment of BnGF14-2c with the flowering-related 14-3-3s in other plant species. AtGF14μ and AtGF14ν are 14-3-3s in *Arabidopsis*; LeGF14-2, LeGF14-5 and LeGF14-74 are 14-3-3s in tomato; OsGF14-c is a 14-3-3 in rice; Gh14-3-3 is a 14-3-3 in cotton. 14-3-3 Proteins contain nine conserved α-helices. (**b**): Homodimer 3D structure modeling of BnGF14-2c. The letter indicated by the red dot is the amino acid corresponding to the start codon, ‘*’ indicates positions which have a single, fully conserved residue, ‘:’ indicates that one of the following ‘strong’ groups is fully conserved.

**Figure 2 plants-11-02312-f002:**
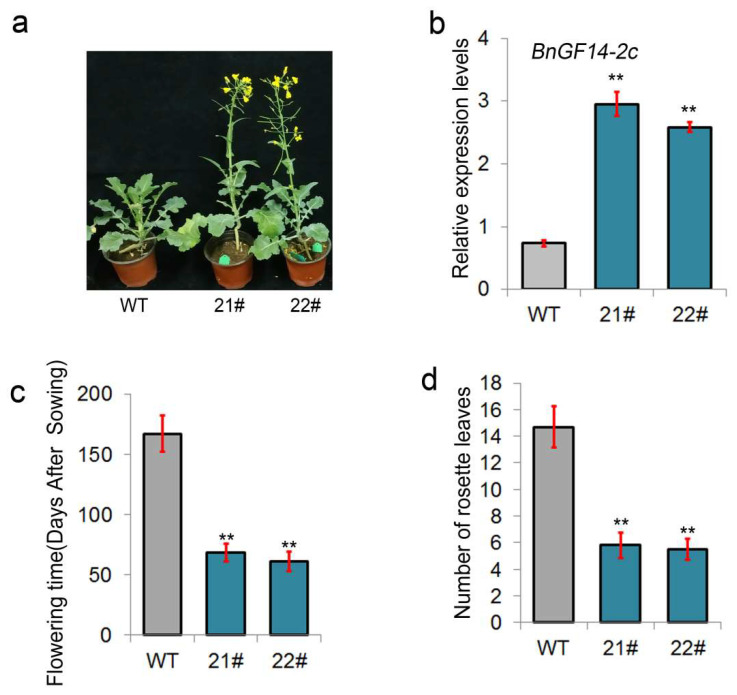
Phenotypic observation and qPCR validation of *35S::BnGF14-2c* transgenic rapeseed. Two positive *BnGF14-2c* transgenic lines (OE21^#^, OE22^#^) and wild type (93275) were selected and analyzed in the greenhouse at 22–24 °C with 55% relative humidity under long-day (LD) conditions (16 h light/8 h dark). (**a**): Phenotypic differences in flowering between 35S::*BnGF14-2c* transgenic plants and wild-type plants. Photographs were taken at 85 day-after-sowing. (**b**): Expression analysis of *BnGF14-2c* by qRT-PCR. (**c**): The statistical analysis of the flowering time (day-after-sowing) in T3 35S::*BnGF14-2c* transgenic plants and wild-type plants. (**d**): The number of rosette leaves in T3 35S::*BnGF14-2c* in transgenic plants and wild-type plants at 85 day-after-sowing. All the experiments were repeated for three times. Asterisk ‘**’ indicates extremely significant differences (*p* < 0.01, Student’s *t*-test) between the transgenic lines and WT.

**Figure 3 plants-11-02312-f003:**
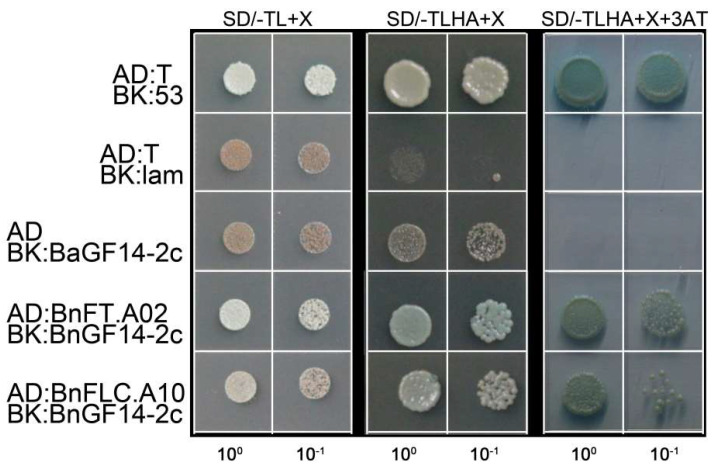
Yeast two−hybrid analysis of the interactions between BnGF14-2c and the candidate interactors. Yeasts expressing the indicated combinations of bait and prey were spotted on the synthetic dropout medium without leucine and tryptophan (SD-Leu/-Trp/X-*α*-gal) and SD medium without leucine, tryptophan, histidine, and adenine (SD-Leu/-Trp/-His/-Ade/X-*α*-gal), both mediums were supplemented with X-*α*-Gal. Yeast strains expressing full length of BnGF14-2c (PGBKT7), BnFT.A02 (PGADT7) and BnFLC.A10 (PGADT7) were indicated in figures, respectively. All the experiments were repeated for three times.

**Figure 4 plants-11-02312-f004:**
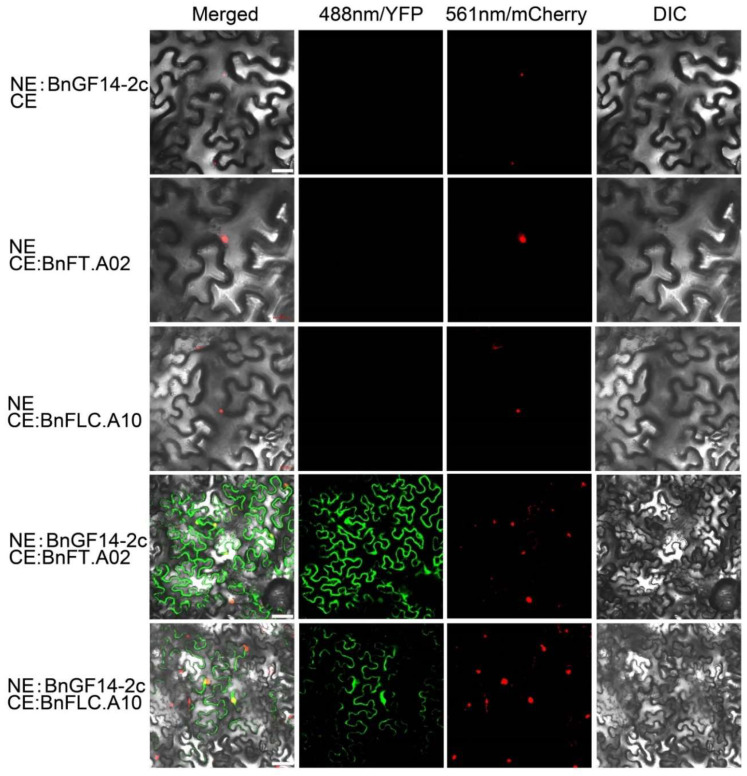
Interactions between BnGF14-2c and the candidates in *N. benthamiana* epidermal cells were tested by BiFC (Bimolecular Fluorescence Complementation). Confocal images were from *N. benthamiana* epidermal cells transiently co-expressing cDNAs encoding nYFP-BnGF14-2c and cYFP-fused BnFT.A02 and BnFLC.A10. NE, blank pSPYNE-YFP; CE, blank pSPYCE-YFP; NE:BnGF14-2c, nYFP-fused BnGF14-2c; CE:BnFT.A02, cYFP-fused BnFT.A02; CE:BnFLC.A10, cYFP-fused BnFLC.A10; Negative control group: nYFP-BnGF14-2c + cYFP, nYFP + cYFP-BnFT.A02 and nYFP + cYFP-BnFLC.A10; experimental group: nYFP-BnGF14-2c + cYFP-BnFT.A02 and nYFP-BnGF14-2c + cYFP-BnFLC.A10; Merged, this diagram was a combination of the three diagrams described below; 488 nm/GFP, GFP fluorescence; 561 nm/mCherry, FIB2-mCherry was specifically localized to the nucleolus as a marker; DIC, bright-field. Scale bar is 25 μm. All the experiments were repeated three times.

**Figure 5 plants-11-02312-f005:**
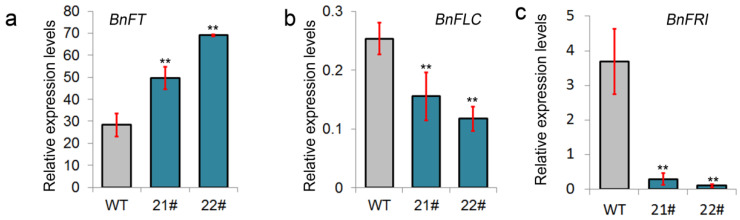
Relative expression levels of some selected genes analyzed using qPCR in BnGF14-2c transgenic lines. (**a**): Expression analysis of *BnFT.A02* by qRT-PCR. (**b**): Expression analysis of *BnFLC.A10* by qRT-PCR. (**c**): Expression analysis of *BnFRI* by qRT-PCR. All the experiments were repeated three times. Asterisk ‘**’ indicates extremely significant differences (*p* < 0.01, Student’s *t*-test) between the transgenic lines and WT.

**Figure 6 plants-11-02312-f006:**
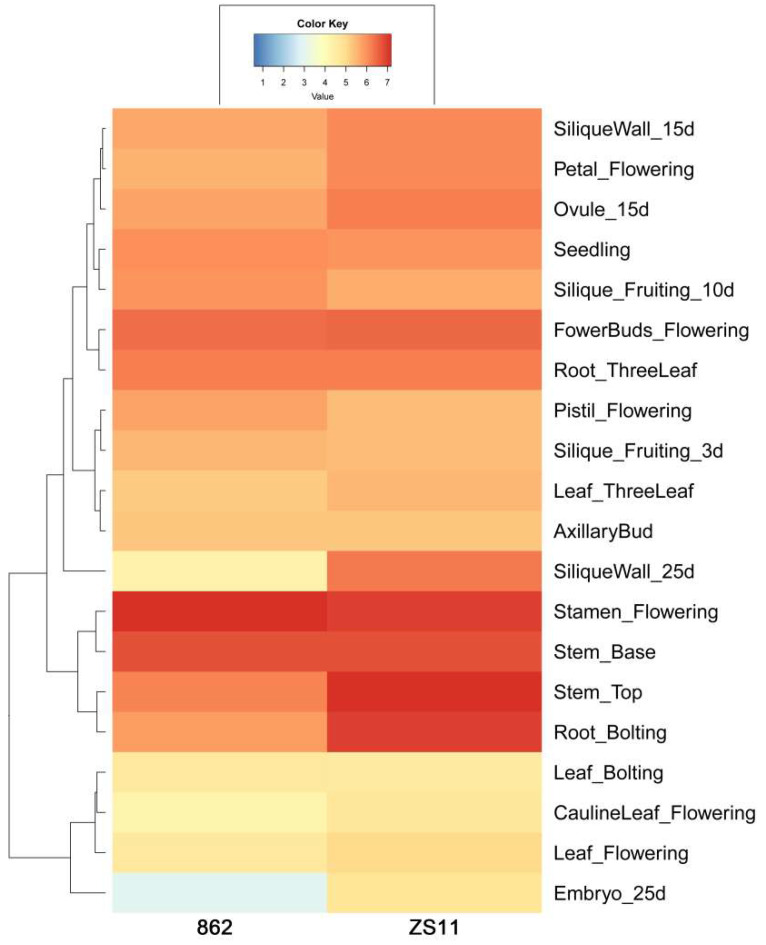
Expression profiles of *BnGF14-2c* across tissues in semi-winter rapeseed ZS11 and spring rapeseed 862. The color scale bar at the top represents log_2_-transformed FPKM values for each gene, with warmer colors denoting higher expression.

**Figure 7 plants-11-02312-f007:**
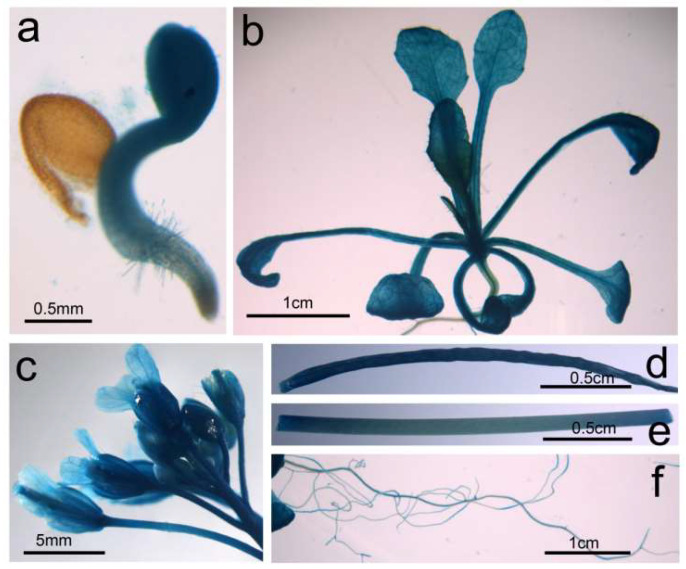
GUS histochemical staining assays of the homozygous *proBnGF14-2c::GUS* transgenic *Arabidopsis*. GUS histochemical assays were detected by using the inverted microscopy (OLYMPUS, MVX10, Japan) with (**a**): 3_d seedling. (**b**): 20_d seedling. (**c**): inflorescences. (**d**): young silique. (**e**): stem. (**f**): young root.

**Figure 8 plants-11-02312-f008:**
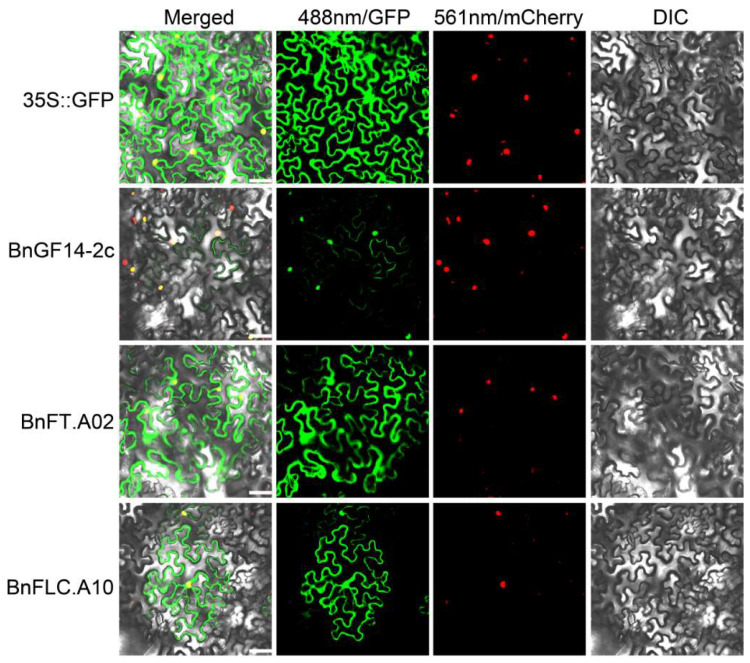
Subcellular localization in *N. benthamiana* epidermal cells. Confocal images were obtained from *N. benthamiana* epidermal cells transiently expressing *BnGF14-2c-GFP*, *BnFT.A02-GFP,* and *BnFLC.A10-GFP*, respectively. 35S::GFP, free GFP signals were located in the cytoplasm. Merged, this micrograph was a combination of the three diagrams described below. 488 nm/GFP, GFP fluorescence. 561 nm/mCherry, FIB2-mCherry specifically localized to the nucleolus was used as a marker. DIC, bright-field. All the experiments were repeated three times.

## Data Availability

BnGF14-2c (BnaA07g34180D), BnFT.A02 (BnaA02g12130D), BnFLC.A10 (BnaA10g22080D), BnFRI (BnaC09g27290D).

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
