# Peer review of "BnGF14-2c Positively Regulates Flowering via the Vernalization Pathway in Semi-Winter Rapeseed"

_plants, 2022, doi:10.3390/plants11172312_

Round 1
Reviewer 1 Report
This paper examines the role of a Brassica napus 14-3-3 protein, BnGF14-2c, in flowering time in a semi-winter variety (able to flower in moderate to no cold) of B. napus. When BnGF14-2c is overexpressed, earlier flowering happens, along with related phenotypes such as reduced rosette leaf number upon flowering. Evidence of direct interaction of BnGF14-2c with genes known to control flowering time and the vernalization response is presented, as is general information about BnGF14-2c expression and subcellular localization. Overall it is a fairly solid effort, though major English usage correction is required. My comments are as follows:
1. In results section 2.1, the phylogenetic analysis leading to the identification of BnGF14-2c as a protein potentially involved in flowering time, is not shown. I believe a phylogeny would be interesting to include, perhaps a tree showing the relationship of BnGF14-2c with other B. napus 14-3-3 genes and other Brassicaceae (especially Arabidopsis) 14-3-3 genes.
2. In Lines 217-218, BnGF14-2c expression is described as being detected in “almost all” tissues. Where is it not detected?
3. The BnGF14-2c subcellular localization studies described in lines 227-236 should be interpreted carefully. A highly expressed transgene can swamp subcellular localization machinery, with the excess localized in the cytoplasm when little might be expressed there. A normal level of expression may result in exclusively or nearly exclusively nuclear localization.
4. Minor comment (Lines 44-45): FRI is not an allele of FLC. Perhaps the authors meant that FRI is an allele of FLA?
5. Minor comment (Lines 63-70) some genes are not described (e.g. TFL) and some gene abbreviations are not spelled out first. It would be good to describe the function of all genes mentioned.
Author Response
Dear Reviewer #1,
Thanks for your timely reply on our submitted manuscript entitled “Over-expression of BnGF14-2c Promotes Flowering in Semi-winter Rapeseed” (Manuscript ID: plants-1771499). We gratefully thank you for making constructive remarks and giving useful suggestions. Each suggested revision and comment, was accurately incorporated and considered, responsed point by point and the revisions are indicated.
Kind regards,
Mr. Fan

Reviewer 2 Report
In this manuscript, author described that Over-expression of BnGF14-2c Promotes Flowering in Semi-winter Rapeseed. In this study, authors identified a homologous 14-3-3 gene BnGF14-2c (AtGRF2_Like in Brassica napus) in rapeseed based on bioinformatic analysis by using the sequences of the flowering-related 14-3-3s in other plant species. Then, the authors found that over-expression of BnGF14-2c in the semi-winter rapeseed “93275” promoted flowering without vernalization. Moreover, both yeast two-hybrid and bimolecular fluorescence complementation analysis indicated that BnGF14-2c could interact with BnFT.A02 and BnFLC.A10, respectively. qPCR analysis showed that the expression of BnFT (AtFT_Like) was increased, and the expression of two selected vernalization-related genes were reduced in the over-expression transgenic plants. Further investigation on subcellular localization demonstrated that BnGF14-2c localized in the nucleus and cytoplasm. The results of RNA-seq analysis and GUS staining indicated that BnGF14-2c is ubiquitously expressed except for the mature seed coat.
Data is solid in this manuscript and written very well. I have found no major flow in this manuscript.
I have a few minor queries for this study.
How did the authors decide the homologous overexpression lines, and why did the author choose only two Ox lines? How many independent lines did the authors generated here?
L122 for flowering to to flower.
L154, L204 . And. There should be no and after full stop.
L188 of in wild type plants to of wild type plants.
Author Response
Dear Reviewer #2,
Thanks for your timely reply on our submitted manuscript entitled “Over-expression of BnGF14-2c Promotes Flowering in Semi-winter Rapeseed” (Manuscript ID: plants-1771499). We gratefully thank you for making constructive remarks and giving useful suggestions. Each suggested revision and comment, was accurately incorporated and considered, responsed point by point and the revisions are indicated.
Kind regards,
Mr. Fan

Reviewer 3 Report
Dear authors,
I had reviewed the paper submitted to Plants journal with title, Over-expression of BnGF14-2c Promotes Flowering in Semi-winter Rapeseed.
The experiments and results had been well designed and presented, respectively. However, the novelty of this work is weak, since the regulatory network of flowering in rapeseed had been well documented (Schiessl, Front Plant Sci. 2020 Nov 20;11:605155. doi: 10.3389/fpls.2020.605155.), and the interaction between FT and GRFs had been demonstrated (Taoka et al., Nature. 2011 Jul 31;476(7360):332-5. doi: 10.1038/nature10272.). Unfortunately, this work needs to be rejected to be published in the Plants journal. A new work for the identification of novel FT-interacted proteins and their functional analysis by using similar experimental strategy in this work is encouraged.
Sincerely,
Author Response
Dear Reviewer #3,
Thanks for your timely reply on our submitted manuscript entitled “Over-expression of BnGF14-2c Promotes Flowering in Semi-winter Rapeseed” (Manuscript ID: plants-1771499). We gratefully thank you for making constructive remarks and giving useful suggestions. Each suggested revision and comment, was accurately incorporated and considered, responsed point by point and the revisions are indicated.
Kind regards,
Mr. Fan

Round 2
Reviewer 3 Report
Dear Authors,
Please check the comments below carefully and revise this work:
1. Provide the experimental results to explain what the role of BnGF14-2c is in the interaction with the flowering promoter, FT, and the flowering suppressor, FLC. Can the interaction of BnGF14-2c and FT or FLC promote or inhibit the function of FT or FLC?
2. Provide the experimental evidences in mRNA and protein levels to explain the function of BnGF14-2c on the flowering genes, including SEP3, FUL, AP1, LFY, AP2, CAL…. and so on.
3. Since the strong promoter, like 35S promoter, will cause false positive, please use the Nos promoter for BiFC. Please provide the results of negative control, cYFP-BnGF14-2c + nYFP, cYFP + nYFP-FT, and cYFP + nYFP-FLC from at least 3 biological replications.
4. Since the authors mentioned the differences between species and the agrobacterium infiltration for Brassica napus are available, in addition to Nicotiana benthamiana, please also provide the results of BiFC from Brassica napus.
Sincerely,
Round 3
Reviewer 3 Report
Dear authors,
Thank you for your responses. Unfortunately, the suggests/comments were not fully and carefully revised, especially no experimental data/results suggested were provided, this manuscript had been decided to reject for publication on the Plants journal.
Sincerely,
Author Response
Dear Reviewer,
Thank you again for your constructive comments and suggestions on this manuscript. In the first round of review, you mainly raised questions about the novelty of this study. In the second round, your suggestions were more specific, suggesting that we should supplement more experimental data and results to enrich the content of the manuscript. We have taken all these into serious consideration and made it clear that there is still a lot of room for expansion in our work. Actually, we have tried our best to reply and explain for your helpful suggestions one by one, and carefully modified the results as well as the wording in the manuscript. So I sincerely hope for your understanding and supporting.
Kind regards,
Mr. Fan